# Seroprevalence and Risk Factors for *Toxoplasma gondii* and *Neospora caninum* in Cattle in Portugal

**DOI:** 10.3390/ani12162080

**Published:** 2022-08-15

**Authors:** Helga Waap, Andrea Bärwald, Telmo Nunes, Gereon Schares

**Affiliations:** 1Laboratório de Parasitologia, Instituto Nacional de Investigação Agrária e Veterinária, Av. da República, Quinta do Marquês, 2780-157 Oeiras, Portugal; 2Centre for Interdisciplinary Research in Animal Health (CIISA), Faculty of Veterinary Medicine, University of Lisbon, 1300-477 Lisboa, Portugal; 3Associate Laboratory for Animal and Veterinary Sciences (AL4AnimalS), 1300-477 Lisbon, Portugal; 4Friedrich-Loeffler-Institute, Institute of Epidemiology, 17493 Greifswald, Insel Riems, Germany

**Keywords:** *Toxoplasma gondii*, *Neospora caninum*, cattle, Portugal, seroprevalence, risk factors

## Abstract

**Simple Summary:**

*Neospora caninum* and *Toxoplasma gondii* are apicomplexan parasites with a worldwide distribution and both infect cattle. While the importance of *N. caninum* is mainly linked to reproductive losses, concern has been raised about the role of beef as a source of food-borne toxoplasmosis in humans. Little is known about the prevalence of these parasites in cattle in Portugal. This study aimed to evaluate the seroprevalence and risk factors for *N. caninum* and *T. gondii* in naturally exposed cattle from different geographical areas in the country. Results point to a low but widespread seroprevalence of *T. gondii*, while that of *N. caninum* was found to be in the same range as reported in other Mediterranean countries. Distribution of both parasites may be explained by common climate factors and management practices identified as significant predictors of infection in the study.

**Abstract:**

*Neospora caninum* has a worldwide economic impact as an important cause of abortion in cattle, while *Toxoplasma gondii*, another abortifacient pathogen, is globally a major foodborne zoonotic threat. The study aimed to evaluate the seroprevalence and risk factors for the two parasites in cattle in Portugal. A total of 612 sera from 35 farms were tested by an in-house p30 ELISA for *T. gondii* and p38 ELISA for *N. caninum. T. gondii* positive and suspicious sera were confirmed by p30 Western blot or IFAT. *T. gondii* and *N. caninum* animal seroprevalence was 9.2% (95%CI 7.1–11.7) and 17.2% (95% CI 14.4–20.4) and herd seroprevalence was 51.4% (95% CI 35.6–67.0) and 68.6% (95% CI 52.0–81.5), respectively. At the univariable level, climate area and precipitation of wettest month, driest month, driest quarter, and warmest quarter were significant predictors of seropositivity for both. *N. caninum* seropositivity was more likely in the region Norte, densely populated areas, and intensive production, and the probability of *T. gondii* seropositivity decreased with herd size. Results confirm the need to consider neosporosis in the differential diagnosis of cattle reproductive disorders in Portugal and may be valuable to inform source attribution models for human toxoplasmosis.

## 1. Introduction

*Neospora caninum* and *Toxoplasma gondii* are obligate intracellular cyst-forming coccidian parasites with a worldwide distribution. Both parasites are capable of infecting a wide range of warm-blooded hosts. *N. caninum* is an important cause of abortion in cattle [1] with a considerable impact on the beef and dairy industries. Economic losses arise mainly from the lower reproductive performance of cattle [1,2] but were also linked to lower milk production, a decrease in weight gain, and premature culling [2,3,4]. *T. gondii* is recognized as a common cause of abortion in small ruminants but does not seem to play a relevant role in reproductive disorders of cattle [5]. Unlike *N. caninum*, whose zoonotic potential has never been clearly elucidated, *T. gondii* is highly ranked among foodborne pathogens. Human toxoplasmosis is particularly dangerous in its congenital form and in people with immunological deficiency. Furthermore, in immunocompetent individuals, where infection is generally considered asymptomatic, ocular toxoplasmosis is currently more often associated with acquired infection than with congenital infection [6,7]. In addition, there has been a growing interest on the influence of *T. gondii* on cognition and mental disorders [8]. Both parasites have a typical heteroxenous apicomplexan lifecycle, with canids and felids acting as definitive hosts for *N. caninum* and *T. gondii*, respectively. Infection of cattle occurs through the ingestion of oocysts in contaminated food and water, while carnivorous and omnivorous hosts may be infected either by the fecal–oral route or by the ingestion of viable tissue cysts in meat. Vertical transmission, another route of infection, may occur through multiple generations in the case of *N. caninum*, being considered the main mode of transmission in cattle herds, but does not seem to play an important role in the case of *T. gondii*. In humans, transplacental infection may occur in non-immune women during pregnancy. In this regard, a European multicenter study estimated that 30–60% of seroconversions in pregnant women are linked to the consumption of raw or undercooked meat while only 6–7% are probably caused by soil contact [9].

*T. gondii* tissue cysts in raw meat can remain viable during several days at common fridge temperatures [10] and survive up to 5 min when heated at temperatures of 60 °C to 100 °C [11].

While the importance of cattle as a source of human infection is generally known [9,12], public health risks through the consumption of beef and cattle-derived products, such as artisan fresh cheese, which was already linked to an outbreak of acute toxoplasmosis in humans, remain unclear [13]. Hence, seroprevalence studies on *T. gondii* in cattle may provide valuable input data for quantitative source attribution models to measure the risk of human infection attributable to the consumption of beef. On the other hand, improving knowledge on *N. caninum* prevalence in the cattle population is important to evaluate the extent and impact of infection on a wider scale and consider this parasite in the differential diagnosis of abortive conditions. In Portugal, little information on the prevalence of *T. gondii* and *N. caninum* in cattle is available, therefore, the aim of this study was to evaluate the presence of specific antibodies against both parasites in naturally exposed animals using a panel of well-characterized sera from different geographical locations.

## 2. Materials and Methods

### 2.1. Sample Characterization

Serological tests were performed on 612 cattle sera selected from a serum collection from a previous cross-sectional study on bovine besnoitiosis in Portugal [14] available at our institute. Blood samples were obtained between 2012 and 2013 using a two-stage cluster sampling scheme, in which a random representative number of farms was selected first and then a random representative number of animals was selected per farm [14]. Sera were stored frozen in 1–1.5 mL aliquots at −20 °C in a temperature-controlled freezer connected to temperature monitoring sensors and thawed for not more than 2–3 times prior to serological testing. For the purpose of this study, a convenience sample was used in which a total of 35 of the above-mentioned farms were selected to cover as much as possible the different geographical areas in Portugal. The number of serum samples available for each farm ranged between 6 and 24. Therefore, for an estimated herd prevalence of 20%, assuming a 90% sensitive test, the number of animals tested per farm should allow identifying at least one positive animal in 32 of the farms with 95% confidence, and with 70–75% confidence in the remaining 3 farms [15]. Animal and farm data (age, sex, herd size, production system and production type, farm location) were available from the former study [14]. All sera were negative for antibodies to *Besnoitia besnoiti* [14].

### 2.2. Testing Strategy

Serological testing for *N. caninum* was performed using an in-house p38 ELISA [16]. Serum samples were tested for antibodies to *T. gondii* by a sequential testing strategy, with an in-house p30 ELISA [17,18] as the first screening assay followed by the p30 Western blot (p30WB) [18,19,20] and Immunofluorescent Antibody Test (IFAT) [21] as confirmatory tests for positive and suspicious results. There is no reference standard to assess seropositivity for *T. gondii* in cattle. Thus, the cut-off for seropositivity by the p30 ELISA was determined based on the empirical 95th percentile method and the cut-off for suspicious sera was set at the 90th percentile, i.e., regarding the 5% strongest reactions as positive and the remaining 10% strongest reactions as suspicious. Confirmation of negative results by the p30 WB and IFAT was performed for a panel of randomly selected negative sera (*n* = 31).

#### 2.2.1. *Neospora caninum p38-ELISA*

The p38 ELISA was carried out as described earlier [16]. The wells of 96-well Polysorb ELISA plates (Nunc) were coated with 120 μL affinity-purified native *N. caninum* p38 tachyzoite surface antigen diluted in 0.1 M bicarbonate buffer (pH 8.3) to a concentration of 0.1 μg/mL and incubated for 1 h at 37 °C. Plates were washed three times and blocked with 300 μL of 20% horse serum in PBS-0.05% Tween (PBS-T) for 30 min at 37 °C. Sera were diluted 1:100 in blocking buffer and distributed in 100 μL duplicates on plates. A positive control diluted 1:80 and a negative control were included in each plate. The positive serum was no longer identical to the serum mentioned in the initial report [16] but came from the same animal. It was stored in aliquots at −20 °C until use, producing identical values as the initial positive control serum, when applied at a 1:80 dilution. After 30 min at 37 °C, plates were washed three times with PBS-T and added with 100 μL monoclonal anti-bovine IgG Clone BG-18 Biotin Conjugate (Sigma) diluted 1:2000 in PBS-T containing 1% horse serum. After 30 min at 37 °C, plates were washed three times, added with 100 μL Extravidin^®^ Peroxidase Conjugate and incubated at 37 °C for another 30 min. The reaction was revealed for 15 min at 37 °C with 100 μL substrate solution prepared with 0.2 M sodium acetate and 0.2 M citric acid containing 100 μg/mL 3,3,5,5-tetramethylbenzidine (TMB) and 0.004% hydrogen peroxide. The colorimetric reaction was stopped with 50 μL of a 2M H_2_SO_4_ solution. Optical densities were read at 450 nm on a Sunrise^TM^ microplate reader (TECAN Deutschland GmbH, Crailsheim, Germany).

#### 2.2.2. *Toxoplasma gondii p30-ELISA*

The p30-ELISA was based on previously established methods [17,18] with few modifications and an adaptation to cattle samples. The wells of 96-well Polysorb ELISA plates (Nunc) were coated with 120 μL affinity-purified native *T. gondii* p30 tachyzoite surface antigen diluted in 0.1 M bicarbonate buffer (pH 8.3) to a concentration of 0.1 μg/mL and incubated for 1 h at 37 °C. Plates were washed three times and blocked with 300 μL of 1% casein in PBS-0.05% Tween for 30 min at 37 °C. One positive and one negative control were included in each plate. Sera and positive and negative controls were diluted 1:100 in blocking buffer and distributed in 100 μL duplicates on plates. As positive control serum, a pool of positive sheep sera was used. Reactivity with the heterologous anti-bovine conjugate was confirmed by Western blot previously (data not shown). The negative control was a serum from a calf reared at Friedrich-Loeffler-Institute and negative for *T. gondii* by IFAT (titer < 1:50). After 30 min incubation at 37 °C, plates were washed three times with PBS-T and developed with 100 μL Peroxidase-conjugated AffiniPure Rabbit Anti-Bovine IgG [H+L] (Jackson ImmunoResearch Laboratories) diluted 1:4000 during 30 min at 37 °C. The colorimetric reaction was stopped and optical densities were read as described for the *N. caninum* p38 ELISA.

#### 2.2.3. *Western Blot*

Western blot analysis to detect antibodies against *T. gondii* p30 protein was performed as described previously [18,19,20] but adapted to cattle samples. Briefly, 100 μL of p30 protein (5 μg/mL) were run in SDS-PAGE on a 12.5% polyacrylamide gel and electro-transferred to polyvinylidene fluoride (PVDF) membranes (Nylonmembran Immobilon-P, Millipore). After blocking with PBS-T/2% gelatin blocking buffer, the blotted membranes were incubated with sera and the same controls used for the ELISA diluted 1:100 in blocking buffer, followed by incubation with peroxidase conjugated AffiniPure rabbit anti-bovine IgG [H+L] (Jackson ImmunoResearch Laboratories (1:500) and developed with a substrate solution prepared with 4-chloro-1-naphthol (Sigma) and hydrogen peroxide.

#### 2.2.4. IFAT

IFAT slides were prepared as described in previous work [21]. Briefly, *T. gondii* tachyzoites were grown in Vero cell (African Green Monkey kidney epithelial cells) cultures using Dulbecco’s Minimal Essential Medium (DMEM) supplemented with 100 UI mL^−1^ penicillin, 100 μg mL−1 streptomycin, 2 mM l-glutamine, and fetal calf serum (10% to initiate cultures and 2% for maintenance) in ventilated 75 cm^2^ cell culture flasks in a CO_2_ incubator at 37 °C. The day before harvesting of tachyzoites, fetal calf serum was withdrawn from the cell culture medium in order to reduce non-specific binding of antibodies. Tachyzoites were purified by filtration through Whatman CF11 cellulose columns [22] and fixed in 4% phosphate-buffered formaldehyde solution. After washing, the antigen suspension was diluted in PBS to a concentration of 2 × 10^6^ tachyzoites mL^−1^ and used to coat IFAT slides. Serum samples were tested for specific anti-*T. gondii* antibodies at 1:50 dilution using anti-bovine IgG (whole molecule) FITC secondary antibody produced in rabbit (Sigma Aldrich, St. Louis, MO, USA) diluted at 1:200 in 0.01% Evans blue solution. Positive and negative sera from previous ELISA and WB analysis were included in each screening round.

### 2.3. Data Analysis

For statistical analysis purposes, an animal was considered seropositive for *T. gondii* if positive or suspicious results by the p30 ELISA were confirmed either by the p30 WB or IFAT and seropositive for *N. caninum* if positive by the p38 ELISA. Apparent animal-level and herd-level prevalence values were calculated with EpiTools Epidemiological Calculator using Wilson’s score for the 95% confidence intervals (CI) [23]. Variables evaluated as potential risk factors included age, herd size, human population density, Csa and Csb Köppen climate areas [24], farm location at NUTSII level, production system (intensive and extensive), production type (dairy and beef), and mean temperature and precipitation values between 1970 and 2000 (bio1–bio19). Climate data were obtained from the Worldclim 2.1 [25] dataset at a resolution of a 1000 m radius from the farm location and human population density (inhabitants/km^2^) was calculated at the municipality level, based on the population census 2011 and the Official Administrative Map of Portugal 2011. For ease of interpretation, the following continuous variables were categorized: age (<24 months and >24 months), herd size (≤50 animals and >50 animals to represent small-scale and large-scale farms), and population density (<50, 50–300, 300–1500, and >1500 inhabitants/km^2^, representing very low, low, semi-dense, and dense population, respectively [26]). Frequency distributions and statistical relationships were analyzed using the statistical software package R 4.2.0 [27]. Statistical significance of associations between serological results and explanatory variables was assessed by a binomial generalized linear mixed model fit by maximum likelihood (Laplace approximation) using the glmer function (lme4 package in R). Both univariate and multivariate analyses were performed. For the univariate analysis, each variable was entered individually into the model, while the multivariate model included all the significant factors from the univariate analysis. The variable “Farm ID” was set as a random factor in both analyses to control for possible variation due to particular characteristics of farms. Statistical significance was set at *p* < 0.05.

### 2.4. Spatial Analysis

Geographical distribution maps were constructed using Quantum Geographic Information System (QGIS) software 3.4. Spatial clustering of *T. gondii* and *N. caninum* seropositive farms was analyzed with the free SaTScan™ software version 9.7. using the Bernoulli probability model for high rates [28]. The following input data were considered: the number of animals with positive and negative results, and the latitude and longitude coordinates of each farm. The model was run with a circular spatial shape window and the default maximum spatial cluster size of 50% of the total population at risk.

## 3. Results

### 3.1. Descriptive Data

The 612 samples included in the survey were obtained in 35 farms located in 31 civil parishes, representing 26 municipalities in four of the five NUTS II regions of continental Portugal: Norte (*n* = 259), Alentejo (*n* = 185), Centro (*n* = 148), and Lisboa (*n* = 20) (Table 1). The 35 herds in the study displayed a median of 102 animals (IQR = 51–211.8), with a minimum of 29 and a maximum of 540 animals. The median age of cattle was 58 months (IQR = 38–87), with a minimum of 9 and a maximum of 240 months. There were 5 sera from male and 607 from female cattle.

### 3.2. N. caninum Seroprevalence Results

Antibodies to *N. caninum* were detected in 105 (17.2%; 95% CI 14.4–20.4) of the sera tested using the p38 ELISA. At farm level, at least one positive animal was detected in 24 farms (68.6%; 95% CI 52.0–81.5%). Positive farms were located in 21 civil parishes (70%) in 17 municipalities (70.8%) included in the study. Animal and herd-level *N. caninum* prevalence according to the categorical variables age, sex, herd size, production system, production type, Köppen climate area, NUTS2 region, and population density considered in the risk factor analysis is shown in Table 1.

### 3.3. T. gondii Seroprevalence Results

Using the p30 ELISA, 31 (5.1%) of the 612 sera were classified as positive at the 95th percentile and further 31 sera were classified as suspicious at the 90th percentile cut-off. Confirmation of p30 ELISA positive and suspicious results either by the p30 WB or IFAT confirmed *T. gondii* seropositivity in 56 animals, representing a seroprevalence of 9.2% (95% CI 7.1–11.7%) (Table 1). Concerning the panel of 31 negative sera, 2 showed a positive reaction by the p30 WB, while none was reactive by IFAT. At farm level, at least one positive animal was detected by the p30 ELISA in 14 farms (40%) when using the 95th percentile as the cut-off for seropositivity and at least one suspicious animal was identified in further 5 farms when using the 90th percentile. A total of 18 farms (51.4%; 95% CI 35.6–67.0%) were found to be positive when confirming positive and suspicious results by the p30WB or IFAT (Table 1). Positive farms were distributed in 17 civil parishes (56.7%) in 14 municipalities (58.3%) included in the study. Distribution of animal and herd-level *T. gondii* prevalence according to the categorical variables age, sex, herd size, human population density, Köppen climate area, NUTS2 region, production system and production type considered in the risk factor analysis is shown in Table 1.

### 3.4. Sera Positive for Both, N. caninum and T. gondii

Of the 105 sera positive for *N. caninum*, 11 were also positive for *T. gondii*. Sera positive for both parasites were from 5 farms. A total of 7 of the *T. gondii* positive sera were confirmed both by p30 WB and IFAT in the 5 farms, 2 sera were confirmed by IFAT only, and other 2 sera by p30 WB only in 3 of the farms.

### 3.5. Univariate Analysis

#### 3.5.1. Risk Factors for *N. caninum* Seroprevalence

Based on the univariate analysis, human population density, Köppen climate area, NUTS2 region, production system, and bioclimatic variables bio13—precipitation of wettest month, bio14—precipitation of driest month, bio17—precipitation of driest quarter, and bio18—precipitation of warmest quarter were significantly associated with *N. caninum* seroprevalence (Table 2). Hence, cattle raised in semi-densely (300–500 inhabitants/km^2^) and densely (>1500 inhabitants/km^2^) populated municipalities were 4.6 (OR = 4.646; 95%CI 1.547–13.949) and 13.64 times (OR = 13.64; 95% CI 2.086–89.176) more likely to be *N. caninum* positive than cattle in areas with a low or very low population density. The odds of infection were higher for cattle raised in Csb climate areas (OR = 4.352; 95% CI 1.411–13.421) compared to Csa areas and higher in the NUTS2 region Norte (OR = 7.11; 95% CI 1.836–27.537) compared to the regions Centro, Lisboa, and Alentejo. In addition, animals raised in intensive production systems were more likely to be positive (OR = 6.396; 95% CI 1.831–22.334), when compared to extensively raised cattle. The odds ratios determined for bioclimatic precipitation variables bio13 (OR = 1.024; 95% CI 1.007–1.041), bio14 (OR = 1.154; 95% CI 1.037–1.283), bio17 (OR = 1.032; CI 1.001–1.057), and bio18 (OR = 1.028; 1.001–1.047) showed a similar positive effect on infection prevalence. The variables age, herd size, and production type (Table 1) were not significantly associated to an increased risk of infection. No statistically significant relationship was found between *N. caninum* and *T. gondii* seropositivity.

#### 3.5.2. Risk Factors for *T. gondii* Seroprevalence

Based on the univariate analysis, herd size, human population density, climate area, and bioclimatic variables bio13—precipitation of wettest month, bio 14—precipitation of driest month, bio17—precipitation of driest quarter, and bio18—precipitation of warmest quarter were identified as significant predictors of *T. gondii* seroprevalence (Table 3). Thus, cattle in large herds (≥50 animals) had a lower risk of *T. gondii* infection (OR = 0.126) compared to animals in small herds, while human population density above 1500 inhabitants/km^2^ increased the odds of infection by 15.786 times (OR = 15.786; 95% CI 1.682–148.139). The probability of being seropositive was higher for cattle in Csb climate areas compared to Csa areas (OR = 4.521; 95% CI 1.269–16.109). The odds ratios determined for bioclimatic precipitation variables bio13 (OR = 1.026; 95% CI 1.007–1.045), bio14 (OR = 1.149; 95% CI 1.022–1.292), bio17 (OR = 1.031; 95% CI 1.004–1.058), and bio18 (OR = 1.025; 95% CI 1.004–1.047) showed a similar positive effect on infection prevalence. The variables age, NUTS2 region, production system, and production type (Table 1) were not significantly associated with an increased risk of infection.

### 3.6. Multivariable Risk Factor Analysis

None of the significant factors in the univariate analysis was associated with a higher risk of infection with *T. gondii* or *N. caninum* in the final multivariate model.

### 3.7. Spatial Analysis Results


*N. caninum*


Spatial cluster analysis identified one significant cluster (*p* < 0.0001) of high occurrence of *N. caninum* infection (Figure 1a). The cluster included a total of 54 positive animals out of 128 in 6 of 7 farms and the resulting cluster radius was 18.7 km. The proportion of positive cases within the cluster area was 42.2% and the relative risk of an individual animal being infected with *N. caninum* compared to animals outside the cluster was 4.0.


*T. gondii*


Spatial cluster analysis identified one significant cluster (*p* < 0.0001) of high occurrence of *T. gondii* infection (Figure 1b)**.** The cluster included a total of 43 positive animals out of 261 in 10 out of 16 farms and the resulting cluster radius was 152.9 km. The proportion of positive cases within the cluster area was 16.5% and the relative risk of an individual animal being infected with *T. gondii* compared to animals outside the cluster was 4.5.

## 4. Discussion

The results of this study show a relatively low but widespread prevalence of *T. gondii* infection in cattle in Portugal. The overall seroprevalence of 8.0% and, in particular, the seroprevalence of 12.0% determined in the NUTS2 region Norte are consistent with previous reports from Portugal (7.5%) [30] and the neighboring Spanish region of Galicia (7.3%) [31]. However, because two of the sera regarded as negative in the ELISA tested positive in one of the tests used for confirmation, it cannot be excluded that the overall *T. gondii* prevalence in Portugal is higher than estimated here. Though similar low proportions of seropositivity (3.2–9.7%) were reported in several surveys in cattle across Europe over the past two decades [32,33,34], exposure to *T. gondii* seems to vary considerably, and other studies point to proportions of seropositive cattle up to 83.3% [35,36,37,38,39]. Regarding seroprevalence results for *N. caninum*, considering only studies in which larger numbers of bovine herds were involved, a higher *N. caninum* seroprevalence in southern Europe compared to northern Europe seems to be apparent (Portugal 28.0% [40], Spain 22.5–22.6% [41], Switzerland 11.5% [42], Belgium 10.2% [43], the Netherlands 9.9% [44], Germany 4.0%, 4.1% or 6.8% [44,45,46], Poland 19.5% [47], Sweden 1.3% or 2.8% [44,48]).

Considering life-cycle characteristics and conditions that might influence the viability of oocysts in soil, differences in the prevalence of both parasites may result from a variety of factors assessed in the present survey, including climate, geographical region, management conditions, herd size, age of animals, human density, or a combination thereof [5,49]. Further, prevalence may also vary according to the type of sample and diagnostic test used.

Regarding climatic factors, weather conditions in continental Portugal are influenced by the latitude and proximity to the Atlantic Ocean and some climatic variables, such as precipitation and temperature, present strong north–south and west–east gradients. According to Köppen’s classification, the climate in Portugal can be grossly divided in two distinct temperate regions: one characterized by rainy winters and dry, hot summers (Csa climate area) in the southern part of the country, and the other by rainy winters and dry, mild summers (Csb climate area) in the north (Figure 1) [29]. In the present study, infection with both parasites was more likely in Csb climate areas, especially in the northwest of Portugal (Table 1, Figure 1). The prevalence of *T. gondii* in soil was shown to be higher in wet or moist seasons characterized by mild temperatures [50] and both sporulation and viability of sporulated oocysts are compromised by extreme heat (>45 °C), and desiccation [51,52,53]. Little is known about the survival of *N. caninum* oocysts in soil, but due to its phylogenetic resemblance with *T. gondii*, a similar resistance to environmental conditions is presumed. Thus, weather conditions in Csb areas in the north of Portugal are expected to be more favorable for oocyst survival across seasons, as opposed to the extreme hot and dry conditions frequently observed during the summer in the Centro-Alentejo area. Further supporting this relationship, when looking more specifically at bioclimatic variables characterizing Csa and Csb climate areas, the ones positively associated with an increased risk of *N. caninum* and *T. gondii* infection were identical and related to precipitation during extreme environmental conditions (wettest and driest month and driest and warmest quarter). In a serological survey on *T. gondii* prevalence in the human Portuguese population comparing three cross-sectional studies spanning three decades (1979–2013), a higher seroprevalence was also observed in the NUTS2 region Norte in the first two decades, although this trend seemed to be inverted in the last decade, with an unexpected raise in the south [54]. The reasons for these contradicting findings, e.g., a change in food habits, with an increase in the consumption or preference for rare beef, still need to be clarified. As for *N. caninum*, the higher infection rates found in the NUTS2 region Norte in the present study (Table 1) reflect previous data in Portugal showing a seroprevalence of 28% in areas of intensive dairy production in the regions Norte and Centro of Portugal [40].

In addition to climate, regional differences in cattle husbandry and management practices are other factors that may contribute to the spatial clustering of *T. gondii* and *N. caninum* infection in the north of Portugal. Thus, in Portugal, dairy production is concentrated in the northwest mainland area, where cows are reared intensively, mostly under permanent confinement conditions. In intensively managed cattle, stocking density and increased potential for contamination of feeding areas, water, fodder, stored silage, and feed supplements by dog feces has been linked to an increased risk of postnatal *N. caninum* infection [49] and could explain the higher prevalence of this parasite in intensive production systems in the present study. On the other hand, the higher internal replacement rates in dairy cattle farms in Portugal compared to beef cattle could also explain the higher prevalence of *N. caninum* through vertical transmission, though the differences for production type (dairy vs. beef) did not reach statistical significance in the GLMM model.

Notably, in a study conducted in cats in France, seroprevalence was increased in areas with high farm densities and during years with cool and moist winters, also showing the effect of climate and farming conditions on the prevalence of *T. gondii* in the definitive host [55]. In the present study, the seroprevalence of both parasites was higher in small-scaled herds, but herd size was found to be a risk factor only for *T. gondii*. Smaller herds are associated to more traditional ways of farming and may favor contamination of feed and water with *T. gondii* oocysts, for instance, through a higher density of cats and lack of biosecurity measures. In fact, studies carried out in several livestock species, including cattle, show that the smaller the herd or flock, the higher the chance for seropositivity to *T. gondii* [5,32,33,37,39]. In the case of *N. caninum*, both risk and protective effects were reported for herd size [49,56].

A possible relationship with indirect risk factors such as human population density, which may play a role in *T. gondii* and *N. caninum* transmission as a result of increased cat and dog populations [53,57], was also assessed in the present survey. Results from this analysis show that the risk of infection was positively associated with increasing population density for both parasites as observed elsewhere [57]. This notwithstanding, *T. gondii* and *N. caninum* transmission dynamics may be influenced by other factors such as heterogeneity in land use and variations in intermediate host and definitive host populations along the urban–rural–wild gradient [58], which need to be addressed in more detail.

Though seroprevalence of *T. gondii* and *N. caninum* is expected to increase with age due to an assumed lifelong persistence of tissue cysts, as noted by several authors [30,33,36,49,59,60], others have demonstrated the opposite [61,62] or that age-prevalence may be influenced by other factors, for instance by the presence of cats in the case of *T. gondii*, with higher seroprevalences in younger animals in farms with cats and in older cows in farms without cats [32]. Here, the lower seroprevalence in animals aged <24 months was not statistically significant. Possible explanations could be the absence of a lifelong antibody response to both parasites in cattle, age-related differences in immune response, or the decrease over time of antibody titers to levels below the detection limit of the serological technique used. Adding to this, cut-off values reported in literature for the serological screening of cattle vary substantially between studies [34,36,49,63,64]. Thus, the presence of low antibody titers in adult cattle could go unnoticed when using more conservative cut-off thresholds in serological surveys. Finally, the serological test itself may have a significant effect on prevalence rates both between and within age groups. Though several tests for the serological diagnosis of *T. gondii* and *N. caninum* were already developed for cattle, few attempts were made to compare their performance in terms of sensitivity and specificity and only a minority of surveys included a second confirmatory test. In the absence of a gold standard test, serial testing approaches may be more advantageous for surveys of diseases with an expected low prevalence, by improving specificity with moderate loss in sensitivity and increasing the positive predictive value of results. Therefore, in regard to *T. gondii*, the strategy in this study was to enhance sensitivity by including a cut-off at the 90th percentile for the p30 ELISA in order to identify suspicious sera, followed by confirmation by two different serological tests to warrant specificity of results. The observation that only a minority of sera were double positive for closely related parasites, *T. gondii* and *N. caninum*, provides confidence that the positive reactions in the *T. gondii* test are true positives and have not been caused by cross-reactions. Interestingly, double positives clustered in a limited number of herds, which shows that specific conditions, probably due to particular epidemiological situations (e.g., presence of dogs and cats) may have led to co-infection.

## 5. Conclusions

Data obtained in the present study may be valuable as input for quantitative source attribution models to assess the likely contribution of beef as a source of *T. gondii* infection in humans. The prevalence determined for *N. caninum* confirms the need to include neosporosis in the differential diagnosis of abortive or reproductive disorders in cattle in Portugal. The risk factors and spatial clustering of infection seen in the present survey for both parasites highlight the importance of biosecurity measures in cattle farms to avoid contamination of feed and water with dog and cat feces, in particular in areas where management and husbandry practices may favor transmission and climate conditions are suited to the survival of oocysts in the environment.

## Figures and Tables

**Figure 1 animals-12-02080-f001:**
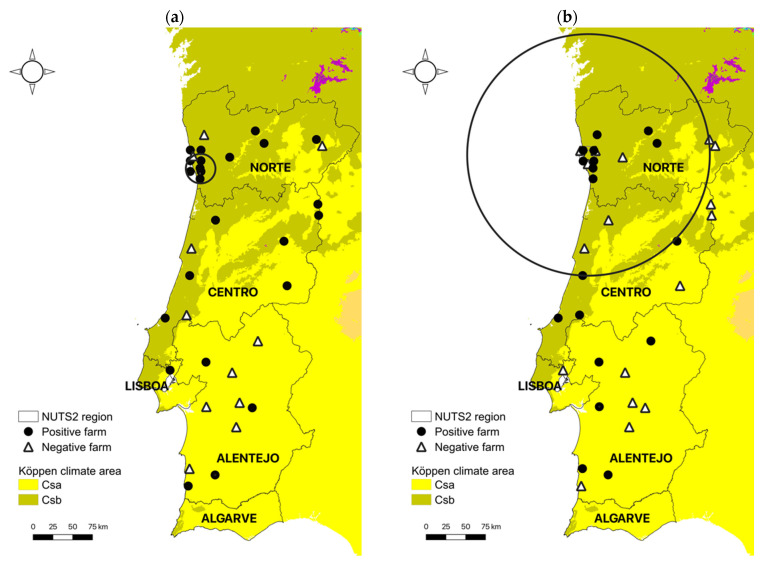
Locations of *Neospora caninum* (**a**) and *Toxoplasma gondii* (**b**) positive and negative farms in the different NUTS2 regions in continental Portugal. Significant clusters of infection detected by spatial cluster analysis represented by circles (small circle with arrow for *N. caninum* and large circle for *T. gondii*). Csa and Csb Köppen climate areas according to Beck et al. 2018 [29]. © EuroGeographics for the administrative boundaries.

**Table 1 animals-12-02080-t001:** Animal and herd seroprevalence (%) of *Toxoplasma gondii* and *Neospora caninum* in cattle in Portugal according to age, sex, herd size, human population density, Köppen climate area, NUTSII regions, production system, and production type. (CI) Confidence Interval for prevalence.

			Animal Seroprevalence					Herd Seroprevalence			
			*T. gondii*		*N. caninum*			*T. gondii*		*N. caninum*	
Variables		*N*	Positive	% (CI)	Positive	% (CI)	*N*	Positive	% (CI)	Positive	% (CI)
Age	≤24	52	4	7.7 (3.0–18.2)	3	5.8 (2.0–15.6)					
	>24	560	52	9.3 (7.2–11.0)	102	18.2 (15.2–21.6)					
Sex	Female	607	55	9.1 (7.0–11.6)	105	17.3 (14.5–20.5)					
	Male	5	1	20.0 (3.6–62.5)	0						
Herd size	≤50 animals	144	34	23.6 (17.4–31.2)	30	20.8 (15.0–28.2)	9	7	77.8 (45.3–93.7)	7	77.8 (45.3–93.7)
	>50 animals	468	22	4.7 (3.1–7.1)	75	16.0 (13.0–19.6)	26	11	42.3 (25.5–61.1)	17	65.4 (46.2–80.6)
Human population density	Very low (<50 inh./km^2)^	220	18	8.2							
	Low (50–300 inh./km^2^)	182	11	6							
	Semi-dense (300–1500 inh/km^2^) inh./km^2^)	173	14	8.1							
	Dense (>1500 inh./km^2^)	37	13	35.1							
Köppen climate area *	Csa	279	11	3.9 (2.2–6.9)	25	9.0 (6.1–12.9)	15	6	40.0 (19.8–64.3)	8	53.3 (30.1–75.2)
	Csb	333	45	13.5 (10.2–17.6)	80	24.0 (19.8–28.9)	20	12	60.0 (38.7–78.1)	16	80.0 (58.4–91.9)
NUTSII Region	Norte	259	37	14.3 (10.6–19.1)	72	27.8 (22.7–33.6)	15	9	60.0 (35.8–80.2)	12	80.0 (54.8–93.0)
	Centro	148	9	6.1 (3.2–11.2)	14	9.5 (5.7–15.3)	9	4	44.4 (18.9–73.3)	7	77.8 (45.3–93.7)
	Lisboa	20	0		3	15.0 (5.2–36.0)	1	0	0.0 (0.0–79.4)	1	100 (20.7–100.0)
	Alentejo	185	10	5.4 (3.0–9.7)	16	8.7 (5.4–13.6)	10	5	50.0 (23.7–76.3)	4	40.0 (16.8–68.7)
Production system **	Intensive	246	34	13.8 (10.1–18.7)	69	28.1 (22.8–33.0)	15	9	60.0 (35.8–80.2)	12	80.0 (54.8–93.0)
	Extensive	282	20	7.1 (4.6–10.7)	23	8.2 (5.5–11.9)	16	7	43.8 (23.1–66.8)	8	50.0 (28.0–72.0)
Production type **	Dairy	209	32	15.3 (11.1–20.8)	56	26.8 (21.3–33.3)	13	9	69.2 (42.4–87.3)	10	76.9 (49.7–91.8)
	Beef	280	19	6.8 (4.4–10.4)	32	11.4 (8.2–15.7)	16	6	37.5 (18.5–61.4)	9	56.3 (33.2–76.9)
	Fattening/Finishing	39	3	7.7 (2.7–20.3)	4	10.3 (4.1–23.6)	2	1	50.0 (9.5–90.6)	1	50.0 (9.5–90.6)
Total		612	56	9.2 (7.1–11.7)	105	17.2 (14.4–20.4)	35	18	51.4 (35.6–67.0)	24	68.6 (52.0–81.5)

* Csa: hot summer Mediterranean climate; Csb: warm summer Mediterranean climate [24]; ** Data for production system and production type available for 528 animals in 31 farms.

**Table 2 animals-12-02080-t002:** Univariate GLMM results for *Neospora caninum* seroprevalence. Farm ID was included as a random effects factor.

Risk Factors	Odds Ratio	Confidence Interval (95%)	*p*-Value
Human population density			
Very low (<50 inhabitants/km^2^)			
Low (50–300 inhabitants/km^2^)			
Semi-dense (300–1500 inhabitants/km^2^)	4.646	1.547–13.949	0.006
Dense (>1500 inhabitants/km^2^)	13.640	2.086–89.176	0.006
Köppen climate area			
Csa			
Csb	4.352	1.411–13.421	0.011
NUTS2 region			
Norte	7.11	1.836–27.537	0.005
Centro			
Lisboa			
Alentejo			
Production system			
Extensive			
Intensive	6.396	1.831–22.334	0.004
Bioclimatic variable			
bio13—Precipitation of wettest month	1.024	1.007–1.041	0.006
bio14—Precipitation of driest month	1.154	1.037–1.283	0.009
bio17—Precipitation of driest quarter	1.032	1.001–1.057	0.01
bio18—Precipitation of warmest quarter	1.028	1.001–1.047	0.004

**Table 3 animals-12-02080-t003:** Univariate GLMM results for *Toxoplasma gondii* seroprevalence. Farm ID was included as a random effects factor.

Risk Factors	Odds Ratio	Confidence Interval (95%)	*p*-Value
Herd size			
≤50			
>50	0.126	0.042–0.377	<0.001
Population density (inhabitants/km^2^)			
<50			
50–300			
300–1500			
>1500	15.786	1.682–148.139	0.0157
Köppen climate area			
Csa			
Csb	4.521	1.269–16.109	0.02
Bioclimatic variables			
bio13—Precipitation of wettest month	1.026	1.007–1.045	0.006
bio14—Precipitation of driest month	1.149	1.022–1.292	0.02
bio17—Precipitation of driest quarter	1.031	1.004–1.058	0.025
bio18—Precipitation of warmest quarter	1.025	1.004–1.047	0.02

## Data Availability

Publicly available datasets were analyzed in this study. Data on cattle population in Portugal can be found at https://dgav.pt, accessed on 2 March 2022. Mean temperature and precipitation values between 1970 and 2000 (bio1–bio19) were obtained from the Worldclim 2.1 dataset available at https://www.worldclim.org/data/worldclim21.html, accessed on 4 October 2021. Human population density at the municipality level (inhabitants/km^2^) was calculated based on census data available at https://censos.ine.pt/xportal/xmain?xpid=CENSOS&xpgid=censos_ficheirosintese, accessed on 2 March 2022, and municipality areas provided in the Official Administrative Map of Portugal CAOP 2011 in https://www.dgterritorio.gov.pt/cartografia/cartografia-tematica/caop, accessed on 2 March 2022. The map in Figure 1 was constructed in QGIS 3.4 Open Source Geographic Information System using freely available layers: administrative boundaries for NUTS2 regions were obtained at https://ec.europa.eu/eurostat/web/gisco/geodata/reference-data/administrative-units-statistical-units, accessed on 2 March 2022, the Köppen-Geiger classification map for climate areas (Beck_KG_V1_present_0p0083.tif) is freely available for download at http://www.gloh2o.org/koppen/, accessed on 24 September 2021.

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
