# Peer review of "Seroprevalence and Risk Factors for Toxoplasma gondii and Neospora caninum in Cattle in Portugal"

_animals, 2022, doi:10.3390/ani12162080_

Round 1

Reviewer 1 Report

Authors performed a retrospective study about parasites seroprevalences. the results obtained are interesting and could provide new insights for portuguese veterinarians and farmes. 

Main comments:

The introduction is well structured and provide an interesting state-of-the-art of the topic. 

M&M. My main issue is about samples. They came from a national bank, so, why did you use so old ones (up to 10 years samples)? Moreover, why do not you study the seroprevalence through years if that samples are available? In this way you could draw an epidemiological map, which could be completely interesting for portuguese authorities, bovine practitioners and farmers.

The main issue is the experimental design of this retrospective study.

Minnor comments:

- ln 74. why over 35 farms? 

- ln 78. If this is a reference, please, follow guidelines. Here and throughout the manuscript.

- ln 90-99. Could you provide some references?

- ln 105, 111, 114, etc. Use the appropriate degree symbol. Here and throughout the manuscript.

- ln 109, 173. Follow the same format for references.

- TABLE 1. There are some lines that cannot be read properly. Why divind herd size in more than 50 or less than 50. WOuld not be interesting as small, medium and large farm size? Explain the difference between herd size and population density? How did you measure it regarding beef and milk farm? Whay is CSA and CSB? And TVOE? Please, include under the table the abbreviations code. In this table I found no p-values, it would be interesting to know.

- Table 2. "residentes" is not english, please translate.

- Figure 1 is veri interesting. I encourage authors to increase the Portugal size, it is not necessary to see that much the iberian peninsula. Moreover, I would like them to use colours in the figure, to make it more attractive.

- ln 346-347. Provide reference.

- Discussion: I would reduce the length of discussion. Moreover, I encourage authors to compare their results with what obtained in another studies in Portugal or some other geographical areas with similar climate, farm conditions, etc. 

Author Response

Comment 1:

The introduction is well structured and provide an interesting state-of-the-art of the topic.

Response: Thank you for the positive appreciation of the introduction section and for all the valuable comments and suggestion for this paper

Comment 2:

M&M. My main issue is about samples. They came from a national bank, so, why did you use so old ones (up to 10 years samples)? Moreover, why do not you study the seroprevalence through years if that samples are available? In this way you could draw an epidemiological map, which could be completely interesting for portuguese authorities, bovine practitioners and farmers. The main issue is the experimental design of this retrospective study.

Response: We agree with the reviewer on the epidemiological value of determining the seroprevalence of T. gondiiand N. caninum along time in Portugal. Unfortunately, at present there is no structure for maintaining a national serum bank that would allow this study in Portugal. The collection of sera utilized in the present work was obtained in the scope of a previous large-scale study on bovine besnoitiosis, in which all regional laboratories agreed to participate. Even though sera are not as recent as desirable, considering the scarce information on bovine toxoplasmosis and neosporosis in Portugal, we believe it was a unique opportunity to assess the seroprevalence of T. gondii and N. caninum in cattle in different farms and geographical locations in Portugal.

Comment 3:

- ln 74. why over 35 farms? 

Response: Many thanks for highlighting that sera were from 35 farms. We decided to change the text to better describe our sampling.
New text: “Blood samples were obtained between 2012 and 2013 using a two-stage cluster sampling scheme, in which a random representative number of farms was selected first and then a random representative number of animals per farm was selected [14]. For the purpose of this study, a convenience sample was used in which a total of 35 of above-mentioned farms were selected to cover as much as possible the different geographical areas in Portugal. The number of serum samples available for each farm ranged between 6 and 24. Therefore, for an estimated herd prevalence of 20%, assuming a 90% sensitive test, the number of animals tested per farm should allow identifying at least one positive animal in 32 of the farms with 95% confidence, and with 70-75% confidence in the remaining 3 farms [15]. “

Comment 4:

 ln 78. If this is a reference, please, follow guidelines. Here and throughout the manuscript.

Response: The link was deleted and included in the data availability statement.

Comment 5:

- ln 90-99. Could you provide some references?

Response: References were added as recommended.

Comment 6:

- ln 105, 111, 114, etc. Use the appropriate degree symbol. Here and throughout the manuscript.

Response: The degree symbol was corrected.

Comment 7:

- ln 109, 173. Follow the same format for references.

Response: The format of the references was corrected.

Comment 8:

- TABLE 1. There are some lines that cannot be read properly. Why divind herd size in more than 50 or less than 50. WOuld not be interesting as small, medium and large farm size? Explain the difference between herd size and population density? How did you measure it regarding beef and milk farm? Whay is CSA and CSB? And TVOE? Please, include under the table the abbreviations code. In this table I found no p-values, it would be interesting to know.

Response: The authors greatly appreciate the reviewer’ suggestions to improve Table 1.  Lines were rearranged as pointed out. Herd size was divided into categories of <50 and >50 animals to represent small-scale and large-scale herds. This information was included in the revised manuscript both in the M&M section under “Data analysis” and in Table 1. Population density refers to the arithmetic human density at the municipality level (inhabitants/km2) and herd size to the number of cattle in the farm. New population density categories based on those defined in OECD et al. (2021) were analyzed during the revision and Table 1 and relevant text sections amended accordingly.  Csa and Csb climate areas definitions were included under Table 1 as recommended. Table 1 was built in order to show seroprevalence results for the different variables analyzed. The statistical analysis to identify risk factors was performed by GLMM analyses and p-values of significant variables are provided in Table 2 and Table 3.

Comment 9:

- Table 2. "residentes" is not english, please translate.

Response: residents was amended to inhabitants

Comment 10:

- Figure 1 is veri interesting. I encourage authors to increase the Portugal size, it is not necessary to see that much the iberian peninsula. Moreover, I would like them to use colours in the figure, to make it more attractive.

Response: Figure one was improved as suggested

Comment 11:

- ln 346-347. Provide reference.

Response: References were included as suggested

Comment 12:

- Discussion: I would reduce the length of discussion. Moreover, I encourage authors to compare their results with what obtained in another studies in Portugal or some other geographical areas with similar climate, farm conditions, etc. 

Response: The discussion was shortened in some paragraphs and the following references were added to compare results in other geographical areas with regards to similar climate and farming conditions.

Panadero R, Painceira A, López C, Vázquez L, Paz A, Díaz P, Dacal V, Cienfuegos S, Fernández G, Lago N, Díez-Baños P, Morrondo P. Seroprevalence of Toxoplasma gondii and Neospora caninum in wild and domestic ruminants sharing pastures in Galicia (Northwest Spain). Res Vet Sci. 2010 Feb;88(1):111-5. doi: 10.1016/j.rvsc.2009.05.010. Epub 2009 May 30. PMID: 19482324.

Afonso E, Germain E, Poulle ML, Ruette S, Devillard S, Say L, Villena I, Aubert D, Gilot-Fromont E. Environmental determinants of spatial and temporal variations in the transmission of Toxoplasma gondii in its definitive hosts. Int J Parasitol Parasites Wildl. 2013 Sep 23;2:278-85. doi: 10.1016/j.ijppaw.2013.09.006. PMID: 24533347; PMCID: PMC3862504.

Reviewer 2 Report

Only one question:

  According author’s opinion, what are the effective and feasible measures to prevent the spread of Toxoplasma gondii and Neospora caninum in cattle in Portugal ?

Author Response

Only one question:

According author’s opinion, what are the effective and feasible measures to prevent the spread of Toxoplasma gondii and Neospora caninum in cattle in Portugal?

Response: Thank you for the review of our manuscript and referring to this important aspect. The importance of biosecurity measures to prevent transmission of T. gondii and N. caninum was included in the conclusion.

Reviewer 3 Report

The authors of the manuscript conducted a seroprevalence study and risk factor analysis for Toxoplasma gondii and Neospora caninum in cattle in Portugal. The survey was well designed, the results were carefully analyzed and presented. The authors' findings are important in the context of veterinary epidemiology and, in the case of T. gondii, also relevant to public health.

I have the following 3 minor points that should be clarified/corrected:

1. Line 58-59: "Though the importance of beef as a source of human infection is generally acquainted [9,12], the role of cattle in the transmission of T. gondii to humans remains uncertain". Please explain this thought in the introduction. You can use the following article - Dubey et al., J Parasitol (2020) 106 (6): 772–788 (https://doi.org/10.1645/20-82).

2. Table 1: Please explain the meaning of the asterisk symbol.

3. Figure 1: The legend is unreadable. Please enlarge it.

Author Response

The authors of the manuscript conducted a seroprevalence study and risk factor analysis forToxoplasma gondii andNeospora caninum in cattle in Portugal. The survey was well designed, the results were carefully analyzed and presented. The authors' findings are important in the context of veterinary epidemiology and, in the case of T. gondii, also relevant to public health.

I have the following 3 minor points that should be clarified/corrected:

Response: Thank you for the careful reading and positive evaluation of the manuscript. We addressed all your comments and suggestions during the revision of the manuscript.

Comment 1:

Line 58-59: "Though the importance of beef as a source of human infection is generally acquainted [9,12], the role of cattle in the transmission of T. gondii to humans remains uncertain". Please explain this thought in the introduction. You can use the following article - Dubey et al., J Parasitol (2020) 106 (6): 772–788 (https://doi.org/10.1645/20-82).

Response: aspects relating to the role of cattle in the transmission of T. gondii to humans were better explained in the introduction section and the suggested reference included

Comment 2:

Table 1: Please explain the meaning of the asterisk symbol.

Response: The asterisk refers to the availability of data for the variables “production system” and “production type” (528 animals in 31 farms). The information was missing in the manuscript and now added at the bottom of the table.

Comment 3:

Figure 1: The legend is unreadable. Please enlarge it.

Response: The legend was enlarged as suggested

Reviewer 4 Report

The manuscript “Seroprevalence and risk factors for Toxoplasma gondii and Neospora caninum in cattle in Portugal” evaluated the seroprevalence and risk factors for T. gondii and N. caninum in cattle from 35 farms of different regions in Portugal. Considering the economic impact of these parasites in herds, it is very important to improve the scientific knowledge in this area. The manuscript is  well structured and I have some doubts and suggestions .

Introduction:

Line 51: phrase “Transplacental infection with T. gondii …” is not linked with the former one.

Materials and Methods:

The authors analyzed sera collected between 2012 and 2013 and these samples were stored for many years. This may be a study bias, so it is important that the authors explain some points such as: storage volume, storage temperature (equipment temperature control), how many times the serum was thawed before performing serological tests. Previous studies of stability of sera for detection of antibodies against other microorganisms indicate good stability of sera stored for long periods for ELISA (10 years or more), but in some cases, the stability for IFAT can be only 4 years. Should the storage time of sera not be taken into account in the discussion of results?     

Line 108: how long was the positive control stored and under what conditions?

Lines 109-110: [16] instead Schares er al., 2000

Line 131: why did the authors use a pool of positive sheep sera for T. gondii as positive control? Do sheep IgG antibodies bind to peroxidase-conjugated anti-bovine IgG?

Line 145: what controls? The same used in ELISA?

Did the authors consider including the presence and number of dogs and cats living at the farms as potential risks? Another potential risk that is statistically significant in other studies is feed storage on the farm. Authors mention these potential risks in the discussion. If the authors have these data, it would be interesting to include them in the study.

Results:

Line 207: what is the explanation for including only 5 male cattle sera in the study? Can the authors say that differences in prevalence found for the variable sex were not significantly associated to an increased risk of infection? (lines 262-264; 287-290)

I suggest that in the item "3.4 Spatial analysis results" the authors describe the result of N. caninum before T. gondii. (just a matter of standardization of the manuscript because in the other items the authors follow this order)

Are the number of positive animals correct in both cases? 43 positive animals for T. gondii in 10 farms? (the other 13 positive animals distributed in 8 farms?) 11 positive animals for N. caninum in 6 farms of the cluster and the other 94 positive animals distributed in 18 farms?

Line 311: (Figure 1b)

Line 318: (Figure 1a)

Figure 1:

Suggestion: increase the size

Describe what the arrow and circles represent

The 35 farms included

I understood that the serum samples studied for both N. caninum and T. gondii are from the same 35 farms. Is it correct? Why are the dots representing farms on maps (“a” and “b”) in different locations?

Discussion:

Line 380: “…geographic areas where….”

Lines 412-413: [57] instead Schares er al., 2004

The authors did not discuss results presented in lines 226-227. Considering that 6,45% of negative sera can be false negative, the seroprevalence would be much higher (14,9%).

Author Response

The manuscript “Seroprevalence and risk factors for Toxoplasma gondii and Neospora caninumin cattle in Portugal” evaluated the seroprevalence and risk factors for T. gondii and N. caninumin cattle from 35 farms of different regions in Portugal. Considering the economic impact of these parasites in herds, it is very important to improve the scientific knowledge in this area. The manuscript is well structured and I have some doubts and suggestions .

Response: The authors would like to thank the reviewer for the thoughtful revision of the manuscript and for all the constructive comments and suggestion which helped to improve the paper

Comment 1:

Introduction:

Line 51: phrase “Transplacental infection with T. gondii …” is not linked with the former one.

Response: The text was modified to link better with the text on vertical transmission.

Comment 2:

The authors analyzed sera collected between 2012 and 2013 and these samples were stored for many years. This may be a study bias, so it is important that the authors explain some points such as: storage volume, storage temperature (equipment temperature control), how many times the serum was thawed before performing serological tests. Previous studies of stability of sera for detection of antibodies against other microorganisms indicate good stability of sera stored for long periods for ELISA (10 years or more), but in some cases, the stability for IFAT can be only 4 years. Should the storage time of sera not be taken into account in the discussion of results?

Response: Thank you for referring to this important aspect. Sera aliquots (1-1.5 ml) were stored frozen at -20 C in temperature-controlled freezers, daily monitored with temperature sensors and thawed maximum 2-3 times before testing. This information was included in the revised manuscript. Regarding storage time, serological testing was performed in 2019, thus storage time was relatively shorter (6-7 years) and less susceptible to affect the stability of sera.

Comment 3:

Line 108: how long was the positive control stored and under what conditions?

Response: Information on storage conditions was added to the manuscript. Modified text:  “The positive serum was no longer identical to the serum mentioned in the initial report (Schares et al., 2000) but came from the same animal and had been aliquoted and stored at -20°C until use.”

Comment 4:

Lines 109-110: [16] instead Schares er al., 2000

Response: The reference was corrected to the right format

Comment 5:

Line 131: why did the authors use a pool of positive sheep sera for T. gondii as positive control? Do sheep IgG antibodies bind to peroxidase-conjugated anti-bovine IgG?

Response: Many thanks for highlighting this. Experimental bovine sera are very rare, thus we decided to use sheep sera. Sera were pooled to have a larger volume and to be able to use positive control not only for this but also future studies. Reactivity of sera, using a heterologous anti-bovine conjugate was confirmed by Western blot analyses. We added further information to the text: “Reactivity with the heterologous anti-bovine conjugate was confirmed by Western blot previously (data not shown).”

Comment 6:

Line 145: what controls? The same used in ELISA?

Response: Yes, the controls were the same used in the ELISA. This information was added to the revised manuscript

Comment 7:

Did the authors consider including the presence and number of dogs and cats living at the farms as potential risks? Another potential risk that is statistically significant in other studies is feed storage on the farm. Authors mention these potential risks in the discussion. If the authors have these data, it would be interesting to include them in the study.

Response: We agree with the reviewer on the interest of assessing the presence of cats and dogs as well as feed storage in farms as potential risk factors. Unfortunately, it was not possible to obtain this information as the sera were collected in the frame of official control programs and these data are not registered during sanitary campaigns.

Comment 8:

Line 207: what is the explanation for including only 5 male cattle sera in the study? Can the authors say that differences in prevalence found for the variable sex were not significantly associated to an increased risk of infection? (lines 262-264; 287-290)

Response: We agree with the reviewer that the number of male sera lacks representativity and removed the variable sex from the statistical analysis.

Comment 9:

I suggest that in the item "3.4 Spatial analysis results" the authors describe the result of N. caninum before T. gondii. (just a matter of standardization of the manuscript because in the other items the authors follow this order)

Response: Thank you for the suggestion. The order of N. caninum and T. gondii spatial analysis results was changed as suggested

Comment 10:

Are the number of positive animals correct in both cases? 43 positive animals for T. gondii in 10 farms? (the other 13 positive animals distributed in 8 farms?) 11 positive animals for N. caninumin 6 farms of the cluster and the other 94 positive animals distributed in 18 farms?

Response: Thank you very much for detecting this mistake. The numbers are correct for T. gondii (43 positives out of 261 = 16.5% in 10 farms), but the number of N. caninum positives was incorrect (corrected to 54 out of 128 = 42.2% in 6 farms)

Comment 11:

Line 311: (Figure 1b)

Response: Identification of figure corrected

Comment 12:

Line 318: (Figure 1a)

Response: Identification of figure corrected

Comment 13:

Figure 1:  Suggestion: increase the size

Response: Size of figure 1 was increased as suggested

Comment 14:

Describe what the arrow and circles represent

Response: A description of arrows and circles was included.

Comment 15:

The 35 farms included 

I understood that the serum samples studied for both N. caninum and T. gondii are from the same 35 farms. Is it correct? Why are the dots representing farms on maps (“a” and “b”) in different locations?

Response: Thank you very much for noticing this imprecision in the maps. The sera tested for T. gondii and N. caninum came from the exactly same farms. We used a point displacement function to make overlapping dots visible in the cluster and by doing so unadvertedly generated maps with dots in apparently different locations. The settings were corrected so that dots/triangles are in the same places in the map.

Comment 16:

Discussion:

Line 380: “…geographic areas where….”

Response: This sentence was deleted in order to shorten the Discussion text as requested by reviewer 1.

Comment 17:

Lines 412-413: [57] instead Schares et al., 2004

Response: corrected

Comment 18.

The authors did not discuss results presented in lines 226-227. Considering that 6,45% of negative sera can be false negative, the seroprevalence would be much higher (14,9%).

Response: Thank you for bringing this point to our attention. In fact, because two of the sera regarded as negative in the ELISA tested positive in one of the tests used for confirmation, it cannot be excluded that the overall T. gondiiprevalence in Portugal is higher than estimated herein. This statement was included in the discussion section of the revised manuscript.

Reviewer 5 Report

General comments: 

The manuscript was well written and this type of research is very exciting. Compliments must be made to the authors for such robust work with respect to samples and analysis of pathogenic protozoa that decrease productivity on dairy farms. There are minor changes to be made which are stated in the specific comments. Once these changes are made this paper should be published.

Specific comments:

The document lacks a Simple summary Section. This must be included.

Materials and Methods

Line 109: Fix referencing to journal format

Line 129: Revise to "Sera, positive and negative controls"

Line 141: revised in text referencing, remove the year.

Line 173: Fix referencing to journal format

An ethical statement should be included. Who granted ethical approval for use of serum or if it was waivered by the institution.

Author Response

The manuscript was well written and this type of research is very exciting. Compliments must be made to the authors for such robust work with respect to samples and analysis of pathogenic protozoa that decrease productivity on dairy farms. There are minor changes to be made which are stated in the specific comments. Once these changes are made this paper should be published.

Response: Thanks a lot for the positive appraisal of the manuscript!  All the corrections based on the reviewer's comments have been performed.

Comment 1:

The document lacks a Simple summary Section. This must be included.

Response: A simple summary was prepared and included in the revised manuscript.

Materials and Methods

Comment 2:

Line 109: Fix referencing to journal format

Response: Done

Comment 3:

Line 129: Revise to "Sera, positive and negative controls"

Response: Done

Comment 4:

Line 141: revised in text referencing, remove the year.

Response: Done

Comment 5:

Line 173: Fix referencing to journal format

Response: Done

Comment 6:

An ethical statement should be included. Who granted ethical approval for use of serum or if it was waivered by the institution.

Response: An ethical statement was included explaining that Ethical review and approval were waived, due to the retrospective character of the study, based on sera selected from a serum collection obtained during a previous survey.

Round 2

Reviewer 1 Report

Authors accomplished all the comments I previously did. I congratulate them for the effort. 

Author Response

Dear Reviewer

Thank you again for all the helpful suggestions that have improved our manuscript.

Reviewer 4 Report

Dear authors, I appreciate the changes made in the manuscpript and I am happy to know that my revision contributed to improve it.

Please, authors should review “Figure 1a” before publication. The small circle with arrow that represents the cluster of farms of high occurrence of N. caninum is different from the Figure in the first version and differs from information in the text (line 336 of 7 farms). It must contain one negative farm.

Author Response

Dear reviewer

We corrected Figure 1 as recommended. Thank you again for all the useful comments.